# Tactile-VLA: Unlocking Vision-Language-Action Model's Physical Knowledge for Tactile Generalization

## Abstract

Vision-Language-Action (VLA) models have shown remarkable achievements, driven by the rich implicit knowledge of their vision-language components. However, achieving generalist robotic agents demands precise grounding into physical interactions, especially in contact-rich scenarios where fine-grained force control is essential. We advance VLAs' implicit knowledge beyond identifying what to do, towards guiding how to physically interact with real world. This paper introduces Tactile-VLA, a novel framework that deeply fuses vision, language, action, and tactile sensing. This framework incorporates a hybrid position-force controller to translate the model's intentions into precise physical actions and a reasoning module that allows the robot to adapt its strategy based on tactile feedback. Experiments demonstrate Tactile-VLA's effectiveness and generalizability in three key aspects: (1) enabling tactile-aware instruction following, (2) utilizing tactile-relevant commonsense, and (3) facilitating adaptive tactile-involved reasoning. A key finding is that the VLM's prior knowledge already contains semantic understanding of physical interaction; by connecting it to the robot's tactile sensors with only a few demonstrations, we can activate this prior knowledge to achieve zero-shot generalization in contact-rich tasks.

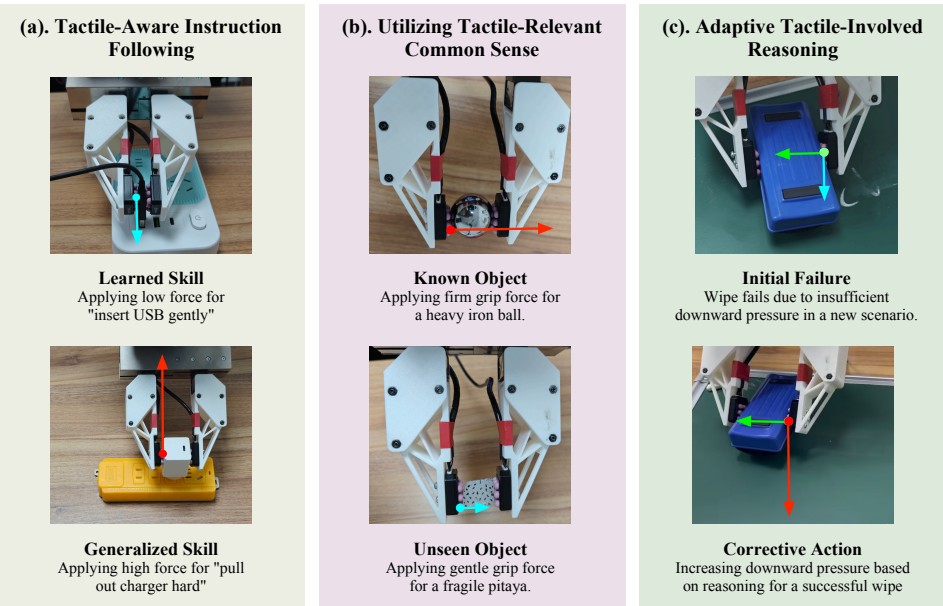

Figure 1: Key capabilities of Tactile-VLA. **(a)** Generalizing language-based force control: It applies force modifiers (e.g., 'gently') learned from one task to a new task for which it only learned the motion. **(b)** Applying implicit common sense: The model automatically uses appropriate firm or gentle grasps for unseen objects without explicit force commands. **(c)** Reasoning to overcome failure: It generalizes reasoning learned from wiping marker ink to autonomously increase force and erase stubborn chalk from a blackboard after an initial failed attempt.

# 1 INTRODUCTION

The development of generalist robotic agents—systems capable of understanding and executing a wide variety of tasks in unstructured environments—has been greatly accelerated by the rise of Vision-Language-Action (VLA) models (Brohan et al., 2022; Zitkovich et al., 2023; Black et al., 2024; Pertsch et al., 2025; Wen et al., 2025). These models leverage pretrained vision-language backbones to interpret abstract instructions and demonstrate impressive generalization abilities across tasks, modalities, and environments. However, while current VLAs excel at high-level reasoning and planning, they often fall short when it comes to grounding their decisions in the fine-grained physical realities of the world, particularly in contact-rich settings where precise force control is critical (Hao et al., 2025; Xue et al., 2025).

A key missing piece in this pursuit is tactile perception. Unlike vision and language, which offer high-level semantic information, tactile sensing provides rich, local, and temporally dynamic feedback about physical interactions—information essential for tasks involving friction, compliance, or material sensitivity (Spelmezan et al., 2009; Sinico et al., 2025). Although some recent efforts have incorporated haptics into robotic frameworks (Pattabiraman et al.; Jones et al., 2025; Yu et al., 2025), haptics is often treated merely as a supplementary perceptual modality, without direct involvement in the policy's action generation.

In this paper, we propose **Tactile-VLA**, a novel method and framework for deeply fusing vision, language, action, and haptics for accomplishing contact-rich manipulation tasks. Moving beyond the current paradigm, Tactile-VLA harnesses the implicit knowledge within vision-language models not just for planning (Yang et al.; Wang et al., 2024; Mei et al., 2024; Hu et al.), but for directly guiding physical interaction at the force control level. This is realized through a hybrid position-force controller that translates the model's learned force targets into precise physical actions, ensuring stability and compliance during contact. This enables a more integrated and intelligent use of language to regulate how actions are performed (e.g. "pick up the apple softly"), not just which actions are chosen (e.g., "pick up the apple"). This fusion of language and haptics supports the emergence of more physically grounded and generalizable robot behaviors. As illustrated in Figure 1, our experiments demonstrate the benefits of this deeper integration across three dimensions: **Tactile-Aware Instruction Following**, which enables robots to learn the meaning of force-related language, such as adverbs like "softly" or "hard", allowing the robot to bridge the gap between abstract intent and physical execution, even in zero-shot scenarios; **Tactile-Relevant Common Sense**, which allows robots to apply world knowledge and semantic reasoning to adjust their contact behavior based on object properties and contextual cues; and **Tactile-Involved Reasoning**, which facilitates feedback-driven control adjustments and autonomous replanning. This is achieved through a Chain-of-Thought (CoT) process where the model reasons over tactile feedback to diagnose failures and formulate corrective actions, especially in the face of novel scenarios or failure cases.

Through Tactile-VLA, we take a step toward tactile-aware generalist agents capable of not only understanding task objectives with semantic intent but also executing them physically with nuanced control and robustness. Overall, our main contributions are threefold:

- We propose **Tactile-VLA**, a novel framework that introduces tactile sensing into VLA models, significantly enhancing semantic grounding and enabling more precise and physically-aware force control in contact-rich tasks.
- We introduce **Tactile-VLA-CoT**, a reasoning-augmented variant that leverages chain-of-thought style interpretation of real-time force feedback to handle task failures and uncertainties, guiding the robot to adaptively replan and adjust its actions during execution.
- We demonstrate that **Tactile-VLA** achieves strong generalization in contact-rich tasks across zero-shot, cross-object, and force-sensitive settings, outperforming standard VLA baselines.

## 2 TACTILE-VLA METHODOLOGY

Tactile-VLA is designed for the fusion of vision, language, haptics, and action modalities to enable more precise and tactile-aware robot manipulation, particularly in contact-rich tasks. This section breaks down the key aspects of the Tactile-VLA framework. We begin by detailing the architecture

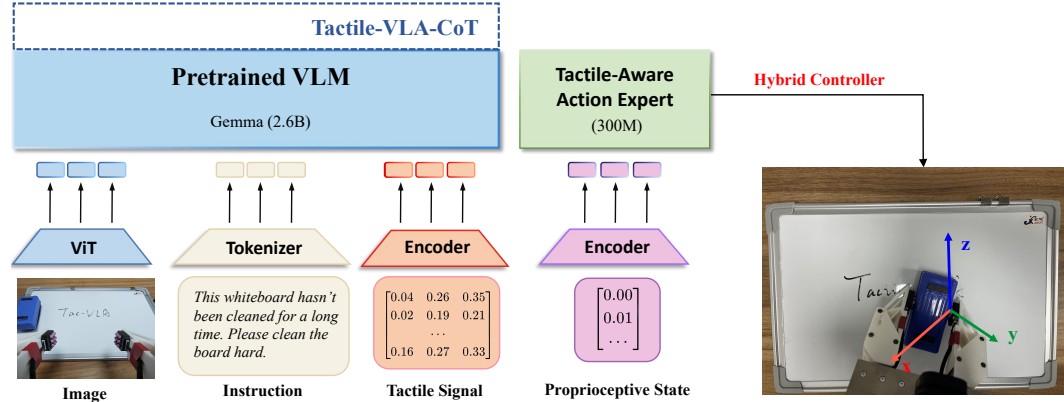

Figure 2: Overview of the Tactile-VLA architecture. Vision, language, tactile, and proprioceptive inputs are separately encoded and fused via a pre-trained Vision-Language Model. The tactile-aware action expert generates target position and force, enabling natural language-guided force control by a hybrid controller and adaptive reasoning in contact-rich manipulation. The dashed block illustrates a CoT-augmented variant, where Chain-of-Thought reasoning enables adaptive motion adjustments based on environmental feedback to handle complex tasks.

and learning process of the policy (Sec. 2.1), followed by the hybrid controller that executes its commands (Sec. 2.2). Then we introduce the CoT-based variant for adaptive reasoning (Sec. 2.3), and conclude with the data collection process that enables the system to learn (Sec. 2.4).

## 2.1 POLICY ARCHITECTURE AND LEARNING

The core design objective of Tactile-VLA is to unlock the physical knowledge inherent in Vision-Language-Action (VLA) models, translating their abstract understanding of interaction into precise, real-world force control. This capability is essential for differentiating commands that share the same motion but differ in force, such as "insert the USB firmly" versus "insert the USB gently". Our model achieves this by creating a direct mapping from multimodal sensory inputs to force-aware action outputs, trained end-to-end with a flow matching objective.

Our architecture employs a token-level fusion approach, deeply integrating multimodal information within the input prefix to the transformer backbone. This design is critical for the advanced reasoning capabilities of Tactile-VLA, particularly for the Chain-of-Thought (CoT) process in the Tactile-VLA-CoT variant (Sec. 2.3). To achieve this, we introduce encoders tailored to the characteristics of each modality. For visual information, we use a pretrained Vision Transformer (ViT) encoder (Dosovitskiy et al., 2020) ($E'_{vis}$), similar to $\pi_0$ (Black et al., 2024), to encode the last $H$ frames into a sequence of distinct token sets. For tactile signals, a simple MLP serves as the encoder $E'_\psi$, which processes the concatenated history of $H$ tactile measurements into a single fused token representing the interaction's temporal dynamics. These resulting visual, tactile, and language tokens are then concatenated to form the unified input prefix sequence $S_t$:

$$S_t = [E'_{vis}(I_{t-H+1}), \ldots, E'_{vis}(I_t), E_{lang}(L_t), E'_\psi([T_{t-H+1}, \ldots, T_t])] \tag{1}$$

where $E_{lang}$ is a common language tokenizer. $S_t$ is then processed by the model's Transformer trunk. A non-causal attention mechanism over this prefix allows the vision, language, and tactile tokens to cross-attend freely, creating a deeply integrated and contextual representation.

This rich representation forms the basis for generating force-aware actions. The prefix is then fed to the **tactile-aware action expert**, which outputs an augmented action vector $a_t$ that explicitly specifies the target position $P_{target}$ and the target contact force $F_{target}$. These targets are provided by the expert demonstrations used for imitation learning. By including force directly in the action space, the model can learn to control the intensity of physical interaction.

The model learns this complex mapping through end-to-end finetuning via imitation learning. The process starts by initializing shared components with pre-trained parameters from $\pi_0$ (Black et al., 2024), a generalist vision-language-action policy. In contrast, newly introduced modules, such as the tactile encoder and the modified action expert, are randomly initialized. The entire model is then

finetuned by employing a Conditional Flow Matching (CFM) objective, where the loss function penalizes deviations in both the kinematic and force dimensions of the predicted action sequence. This learning mechanism is what compels the model to leverage the VLM's latent physical knowledge, ultimately creating a direct mapping between linguistic nuances (e.g., "gently") and their corresponding physical force magnitudes (e.g., $0.5N$).

## 2.2 HYBRID POSITION-FORCE CONTROLLER

Once the tactile-aware action expert determines the target position and target force, a low-level controller is required to balance these two distinct objectives. Our strategy is position-dominant and ultimately realized through position commands, acknowledging that most manipulation tasks are dominated by precise kinematic motion, with force control required merely during contact phases (Raibert & Craig, 1981). To integrate force objectives, we adopt an indirect force control method inspired by impedance control principles (Hogan, 1985). This involves translating force targets into adaptive adjustments of the position command.

However, unlike classic impedance control which aims for passive compliance, our objective is the active tracking of a target force. The controller measures the force error $\Delta F = F_{\text{target}} - F_{\text{measured}}$, which is used to compute a corrective positional adjustment only when its magnitude $\|\Delta F\|$ exceeds a predefined threshold $\tau$ to enhance operational smoothness:

$$P_{\text{hybird}} = P_{\text{target}} + \begin{cases} K \cdot \Delta F & \text{if } \|\Delta F\| > \tau \\ \mathbf{0} & \text{if } \|\Delta F\| \leq \tau \end{cases} \quad (2)$$

where $K$ is a gain matrix. A Proportional-Integral-Derivative (Willis, 1999) controller then actuates the robot's joints to the dynamically updated $P_{\text{hybird}}$. Specifically, we decouple the control of two distinct force components: the net external force and the internal grasping force. The key principle of this separation is to establish two independent control channels. The gripper's Cartesian position is used to exclusively regulate the net external force applied to an object, while the gripper width is used in parallel to control the internal grasping force, thus dictating how firmly the object is held.

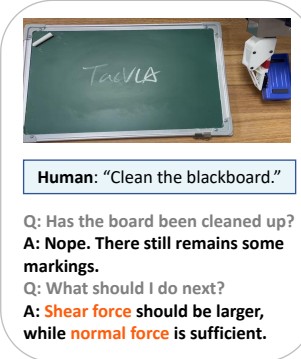

**Human**: "Clean the blackboard."

Q: Has the board been cleaned up?
**A: Nope. There still remains some markings.**
Q: What should I do next?
**A: Shear force should be larger, while normal force is sufficient.**

Figure 3: The working process of Tactile-VLA-CoT on Wiping the Board task.

## 2.3 TACTILE-VLA-CoT: REASONING-BASED ADAPTATION

While the core Tactile-VLA architecture provides fine-grained force control, leveraging its inherent reasoning capabilities is key to further unlocking VLM's potential for robust adaptation (Stone et al.; Huang et al., 2023; Shi et al., 2024; Belkhale et al., 2024). To this end, we propose Tactile-VLA-CoT, a variant that integrates Chain-of-Thought (CoT) to activate and utilize the VLM's latent reasoning skills (Wei et al., 2022; Chen et al., 2024; Zhang et al., 2024; Lin et al., 2025). In this variant, force and tactile feedback are treated as more than just policy inputs; they become crucial cues for adaptive reasoning and re-planning.

The CoT process is realized by using VLM's own pretrained decoder to generate an explicit internal monologue. This monologue allows the model to reason about the cause of a failure, such as an unexpected slip, and formulate a corrective action. To enable this, we finetune the model with a small, targeted dataset of demonstrations. Each sample in this dataset captures a specific failure event (e.g., wiping a blackboard with slippage) and pairs the multimodal sensory stream with a language annotation analyzing the failure's cause. This training serves a dual purpose: first, it preserves the VLM's general reasoning abilities, mitigating catastrophic forgetting during finetuning. More importantly, it extends this reasoning to the tactile modality, teaching the model to infer physical phenomena from sensor signals, such as detecting insufficient downward pressure when wiping or tool slippage from shear force signals.

In practice, this CoT reasoning is triggered at fixed intervals. This simple and effective approach allows the model to periodically review its progress. The prompt structure first requires the model to determine if the task was successfully done. If deemed a failure, the model is prompted to analyze the underlying causes using the sensory feedback, as shown in Figure 3. The resulting reasoning

output explicitly analyzes different force components (e.g., "grasping force is sufficient, but normal force is too low") and then formulates a new, corrective instruction to guide the next attempt, for example, generating "wipe the board again, but apply more downward force." This process enhances the system's ability to handle complex scenarios by making the adaptation process explicit and grounded in physical interaction.

## 2.4 DATA COLLECTION

Accurate and semantically aligned tactile data is critical for training agents in contact-rich scenarios. Conventional teleoperation is insufficient for this purpose, as the human operator typically lacks direct force feedback. A policy collected this way would inherently not depend on tactile information, rendering it unsuited to the learning objective. To address this, we constructed a specialized data collection setup by building upon the Universal Manipulation Interface (UMI) (Chi et al., 2024), a portable, handheld device. We augmented the UMI gripper with dual high-resolution tactile sensors, capable of capturing both normal and shear forces, allowing operators to directly sense contact dynamics and provide demonstrations that are explicitly guided by force. Details of temporal synchronization and so on are illustrated in Appendix D.

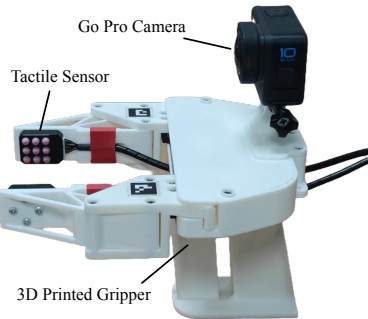

Figure 4: Data collection setup

## 3 EXPERIMENT

In this section, we investigate the effectiveness of our proposed Tactile-VLA model on different tasks. Specifically, we conduct experiments on several contact-rich manipulation scenarios, which require multi-modal perception including vision, language, and haptics. The goal of our experiments is to answer three research questions: **RQ1:** How effectively can Tactile-VLA interpret and generalize abstract, force-related language commands across different contact-rich tasks? (Sec. 3.2) **RQ2:** To what degree can the model leverage the VLM's inherent common-sense knowledge to infer and apply appropriate interaction forces for unfamiliar objects? (Sec. 3.3) **RQ3:** Does the integration of tactile feedback enable the model to reason about physical failures and autonomously adapt its force-based strategy to ensure task success? (Sec. 3.4)

### 3.1 IMPLEMENTATION DETAILS

**Baselines.** To answer the above questions, we compare the following baseline methods and ablation methods with the proposed Tactile-VLA on various tasks: $\pi_0$-base, a Vision-Language-Action flow model for general robot control; $\pi_0$-fast, a variant of $\pi_0$-base; Tactile-VLA, our method; and Tactile-VLA-CoT, a variant of Tactile-VLA with a CoT reasoning process.

**Tasks and Data Collections.** We mainly focus on three contact-rich manipulation tasks, as visualized in Figure 5 and Figure 7: Charger/USB Insertion and Extraction, Tabletop Grasping, and Wiping the Board. In **Charger/USB Insertion and Extraction**, the robot must pull out a charger or USB and plug it into the correct socket. For the training data, we collected 100 demonstrations each for "soft" and "hard" USB manipulations, and another 100 demonstrations for the charger task to learn the basic motion. In **Tabletop Grasping**, the robot is required to grasp various objects with an appropriate force, judging in advance whether they are heavy or fragile. This task was trained using 50 demonstrations for each object. Six objects visualized in Figure 5 could be seen in the training phase, while an additional six unseen objects are introduced for evaluation. In the **Wiping the Board** task, the robot is expected to wipe a board with a default force, evaluate the result, and then adjust the force as needed. To enable this reasoning, the training data consists of 100 successful and 100 failed wiping demonstrations on a whiteboard, while the model has never encountered wiping on a blackboard during training.

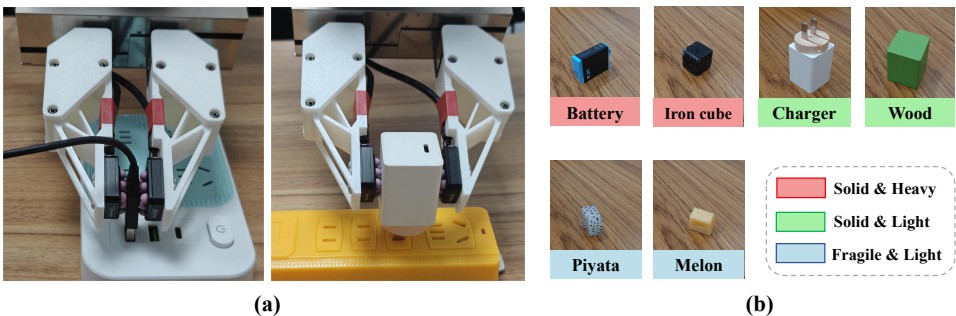

Figure 5: (a) The charger insertion and extraction task. (b) A selection of objects for the tabletop grasping task, only including in-domain (ID) items, categorized by their physical properties.

## 3.2 TACTILE-RELEVANT INSTRUCTION FOLLOWING

This experiment is designed to evaluate a core hypothesis of our work: whether `Tactile-VLA` can learn a generalizable understanding of force-related adverbs (e.g., "softly", "hard") from one task and apply that semantic knowledge to a different, unseen task. Specifically, we investigate if the model, after being trained to associate "softly" and "hard" with specific force profiles in a USB insertion task (Task A), can successfully transfer this understanding to a charger insertion task (Task B) for which it has only learned the motion but received no corresponding linguistic force commands. This tests for true semantic grounding, where language directly modulates physical interaction in a zero-shot context.

We define two distinct but kinematically similar contact-rich tasks, which are visualized in Figure 5(a):

- **Task A (USB Insertion and Extraction):** The robot is trained on demonstrations of pulling out a USB cable and re-inserting it into another socket. The training data for this task is augmented with explicit, force-related natural language instructions, such as, "pull out and insert the USB softly into the left socket".

- **Task B (Charger Insertion and Extraction):** The robot learns to pull out a power charger and plug it into a power strip. Crucially, the expert demonstrations for this task contain only the kinematic motions; no language instructions related to force ("softly"/"hard") are provided during its training phase.

We compare the performance of our `Tactile-VLA` against two baselines, $\pi_0$ and $\pi_0$-fast, which lack our tactile-fusion architecture. Evaluation is based on two key metrics: (1) **Success Rate (%)** for both tasks to measure overall robustness and precision, and (2) **Applied Insertion Force (N)** to quantify how the models interpret adverbial commands during the charger insertion task, for which they were not explicitly trained.

**Results and Analysis.** Our results, presented in Table 1 and Table 2, demonstrate `Tactile-VLA`'s superior performance and generalization capability. As shown in Table 1, `Tactile-VLA` achieves a significantly higher success rate than both baselines across the two tasks. We attribute this to the deep fusion of tactile feedback, which allows for more precise and adaptive control during the critical contact-rich phases of insertion, reducing failures from misalignment or excessive force.

Table 1: Success rates on USB/Charger insertion and extraction tasks.

| Model | USB (%) | Charger (%) |
|---|---|---|
| $\pi_0$-base | 5 | 40 |
| $\pi_0$-fast | 0 | 25 |
| Tactile-VLA | 35 | 90 |

More importantly, Table 2 provides direct evidence for rich semantic generalization. For the learned task, our model correctly applied distinct forces corresponding to the explicitly trained words "softly" (0.51N) and "hard" (2.57N). It also successfully generalized within the same task to a spectrum of unseen but related adverbs, correctly inferring intermediate forces for commands like "gently" (0.75N) and "firmly" (1.98N).

Table 2: Applied force (N) under different instructions.

| Model | Learned Task (USB) | | | | | | Generalized Task (Charger) | |
|---|---|---|---|---|---|---|---|---|
| | Learned Words | | Generalized Words | | | | Zero-shot | |
| | 'softly' | 'hard' | 'gently' | 'firmly' | 'rigidly' | 'harder' | 'softly' | 'hard' |
| $\pi_0$ | 2.41 | 2.68 | 2.35 | 2.72 | 2.53 | 2.29 | 6.61 | 5.69 |
| $\pi_0$-fast | 2.61 | 2.33 | 2.79 | 2.45 | 2.26 | 2.58 | 7.37 | 6.42 |
| Tactile-VLA | 0.51 | 2.57 | 0.75 | 1.98 | 2.42 | 2.94 | 4.68 | 9.13 |

Even more impressively, the model demonstrated an ability to extrapolate beyond the bounds of its training data, applying 2.94N for the novel command "harder"—a force greater than that for the trained "hard" command. This understanding was effectively transferred in the zero-shot generalized task, where our model applied a strong 9.13N force for 'hard' and a significantly lower 4.68N for 'softly'. This demonstrates a learned, generalizable, cross-modal understanding of force-related language. In stark contrast, the $\pi_0$-base and $\pi_0$-fast baselines, lacking the mechanism to ground this language in physical force, failed to differentiate their applied force; as shown in the table, their force application shows no correlation with the adverbial commands across all conditions. This highlights our model's ability to bridge the gap between abstract language and nuanced physical execution, a key advancement for creating more intelligent and versatile robotic agents.

## 3.3 TACTILE-RELEVANT COMMON SENSE

In real-world manipulation tasks, robots must exhibit the ability to generalize prior knowledge across modalities. In particular, the integration of prior knowledge from a VLM into haptic signals is critical for effective manipulation. For instance, the robot must adapt its grasp by reasoning about an object's properties, applying different magnitudes of force for distinct categories: a firm force for **Solid & Heavy** objects, a moderate force for **Solid & Light** objects, and a gentle force for **Fragile & Light** objects to prevent damage. This capacity to adapt the applied force based on prior visual and contextual knowledge is essential for performing a variety of manipulation tasks effectively.

Table 3: Success rates (%) for grasping various objects without causing deformation. The models are evaluated on in-domain (ID) and out-of-domain (OOD) objects, with training primarily focused on medium-stiffness items. Rates are calculated from 10 trials per object.

| Model | Solid & Heavy Objects | | | | Solid & Light Objects | | | | Fragile & Light Objects | | | |
|---|---|---|---|---|---|---|---|---|---|---|---|---|
| | ID | | OOD | | ID | | OOD | | ID | | OOD | |
| | Iron cube | Battery | Nail | Steel Ball | Wood block | Charger | Plastic | Toy | Pitaya | Melon | BlueBerry | PaperBox |
| $\pi_0$-base | 100 | 80 | 30 | 60 | 60 | 70 | 40 | 30 | 50 | 0 | 0 | 0 |
| $\pi_0$-fast | 70 | 60 | 10 | 70 | 70 | 50 | 30 | 40 | 40 | 10 | 0 | 0 |
| Tactile-VLA | 100 | 90 | 100 | 90 | 90 | 100 | 80 | 90 | 90 | 80 | 100 | 90 |

During training, the robot learns the appropriate grasping force for each object category. A selection of these in-domain objects, categorized by their properties, is shown in Figure 5(b). A grasp is considered **successful** if the object is securely lifted in a single attempt without observable deformation. Notably, even within these more specific categories, the precise force required may still vary between individual objects. Once the robot has learned the grasping forces for these objects, we proceed with evaluations on both *in-domain* and *out-of-domain* objects. These evaluations test the robot's ability to generalize across a range of object types not encountered during training.

**Results and Analysis.** The results demonstrate that `Tactile-VLA` successfully learns the haptic information corresponding to in-domain objects, while also exhibiting strong generalization to out-of-domain objects. As shown in Table 3, our model achieves substantially higher success rates in grasping both in-domain and out-of-domain objects compared to baselines, especially for fragile items where it succeeds without causing damage. Furthermore, Figure 6 shows that `Tactile-VLA` correctly infers the appropriate interaction force, applying hard, medium, or soft grasps to heavy, light, and fragile objects respectively, even for those it has never seen before. This finding provides compelling evidence that our model successfully transfers knowledge from the VLM to the tactile modality, endowing the robot with strong generalization capabilities. Rather than merely fitting to training data, the model appears to leverage prior knowledge to handle a broader range of scenarios, demonstrating its potential for real-world applications in contact-rich manipulation tasks.

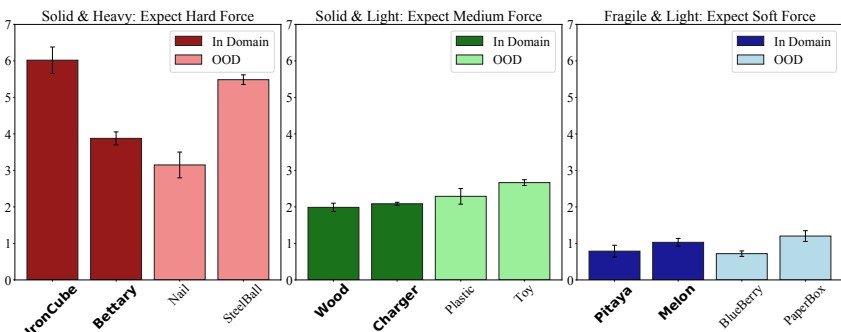

Figure 6: Applied grasping force for various objects, categorized by hardness and whether they are in-domain (ID) or out-of-domain (OOD). Each bar represents the mean applied force over 5 trials, with error bars indicating the standard deviation.

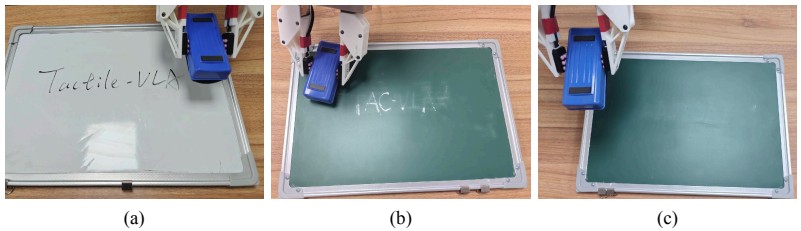

| (a) | (b) | (c) |

Figure 7: Generalizing Wiping Strategies through Tactile-Involved Reasoning. (a) The model is first trained to wipe marker ink from a whiteboard. (b) In a zero-shot transfer to a novel blackboard task, this initial policy fails as the applied force is too low for chalk. (c) By reasoning over the physical feedback of the failure, the model adapts its strategy, increases the applied force, and successfully completes the task.

## 3.4 TACTILE-INVOLVED REASONING

To validate our model's capacity for adaptive reasoning, we designed an experiment to specifically test its ability to interpret physical feedback and autonomously adjust its strategy. This moves beyond merely following instructions to demonstrating an understanding of task success or failure through tactile interaction, a key claim of our work. We investigate whether `Tactile-VLA-CoT` can generalize a learned reasoning process from a familiar task (wiping a whiteboard) to a novel, physically distinct scenario (wiping a blackboard), which requires a different level of force, as illustrated in Figure 7.

The experiment is centered on a "Wipe the Board" task, structured to assess adaptive reasoning. In the training phase, data is collected on a whiteboard. This dataset contains a mix of successful demonstrations and various failure cases. For instance, some demonstrations feature insufficient force, failing to erase the marker. These failure cases are paired with supervisory text that articulates a corrective thought process (e.g., *"The force was too light. A stronger force is needed. Now trying with 5N."*) to train the reasoning module of `Tactile-VLA-CoT`. Successful demonstrations using higher, appropriate forces are also included, reinforcing the connection between the correct force and task success. Subsequently, in a zero-shot generalization test, the robot is presented with a blackboard—a novel object whose chalk markings require significantly more force to erase. The robot is instructed simply to *"Wipe the board."* After an initial attempt with a default force proves insufficient, the `Tactile-VLA-CoT` model is expected to autonomously trigger its Chain-of-Thought reasoning as shown in Figure 3. This allows it to analyze the tactile feedback and adapt its action plan by increasing the applied force, without any prior training on blackboards.

**Results and Analysis.** As shown in Table 4, our results show that `Tactile-VLA-CoT` successfully generalizes its reasoning capabilities to the novel blackboard task. In the zero-shot test, the robot initially attempted to wipe the chalk with a default force of 3.5 N, which failed. Recognizing the lack of progress from physical feedback, the CoT module generated a chain of reasoning concluding that greater force was required.

Subsequently, the model autonomously increased the applied force to 6.7 N—a level 34% greater than the 5 N force associated with the attempts in the whiteboard training data. This adaptation was sufficient to successfully erase the chalk mark. On the original whiteboard task, `Tactile-VLA` achieved a high success rate, significantly outperforming the baseline VLA. In the more challenging zero-shot black-

Table 4: Success rate over ID and OOD scenarios.

| Type | In Domain (Whiteboard) | Out of Domain (Blackboard) |
|---|---|---|
| $\pi_0$-base | 40 | 0 |
| $\pi_0$-fast | 45 | 0 |
| Tactile-VLA | 80 | 15 |
| Tactile-VLA-CoT | 75 | 80 |

board scenario, the distinction was stark: our model succeeded through reasoning-based force adaptation, whereas the baseline model failed completely. Although the baseline could replicate the wiping motion, it lacked the mechanism to interpret the tactile failure and could not increase its force, repeatedly executing the same ineffective, low-force action. This highlights the critical role of tactile-centered reasoning in achieving robust and generalizable manipulation in contact-rich tasks.

## 4 RELATED WORK

**Vision-Language-Action (VLA) Models.** The advent of large-scale VLA models has transformed robot manipulation. Influential works such as RT-1 (Brohan et al., 2022), RT-2 (Zitkovich et al., 2023), Octo (Team et al., 2024), $\pi_0$ (Black et al., 2024), VIMA (Jiang et al.), PALM-E (Driess et al., 2023), OpenVLA (Kim et al.), and Gato (Reed et al.) have demonstrated unprecedented generalization by mapping vision and language inputs to action sequences. While effective, VLAs that rely mainly on visual feedback face limitations in contact-rich tasks where vision can be occluded or ambiguous. Recent work thus explores integrating tactile and force sensing into the VLA paradigm, with examples including concurrent works like FuSe (Jones et al., 2025), ForceVLA (Yu et al., 2025), and other vision-tactile-language policies (Lin et al., 2024; Huang et al.). Unlike concurrent works that focus on finetuning with auxiliary losses (Jones et al., 2025) or modality-specific routing (Yu et al., 2025), Tactile-VLA's primary contribution is demonstrating that a VLM's latent space already contains a rich, semantic understanding of physical interaction; by directly connecting this to tactile sensors with only a few demonstrations, we unlock this prior knowledge to achieve zero-shot generalization in contact-rich tasks.

**Tactile Integration in Robot Policies.** Beyond the VLA paradigm, extensive research has explored integrating tactile signals into robot policies. The technical strategies are diverse, ranging from classic control methods to modern learning-based policies for tasks like grasping (Calandra et al., 2018; Polic et al., 2019), insertion (Dong et al., 2021; Ma et al., 2019), in-hand manipulation (She et al., 2021; Qi et al., 2023), fabric handling (Sunil et al., 2023), and tool use (Wang et al., 2021). These efforts have produced a variety of effective, specialized policies. Among learning-based approaches, strategies such as hierarchical architectures that decouple planning and control (Xue et al., 2025), reinforcement learning with shaped rewards (Schoettler et al., 2020), force-centric imitation learning (Liu et al., 2024), and end-to-end visuo-tactile policies (Yu et al.) have been developed. While these specialized policies perform well on their target tasks, the lack of language limits their ability to follow novel instructions, reason about abstract goals, or use commonsense. Our work aims to combine the physical precision of these tactile-informed policies with the semantic flexibility and broad world knowledge of modern VLAs.

## 5 CONCLUSION

This paper introduced Tactile-VLA, a framework built on the fundamental finding that Vision-Language-Action models (VLMs) possess a latent, semantic understanding of physical interaction that can be unlocked for complex, contact-rich tasks. Our core contribution is an architecture that deeply fuses tactile sensing as a native modality, creating the essential bridge between the VLM's abstract knowledge and the dynamics of physical force. By connecting the VLM to tactile sensors with only a few demonstrations, we have shown it is possible to unlock this powerful prior knowledge to achieve zero-shot generalization in tasks requiring nuanced physical interaction.

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

## A  TASKS AND EVALUATIONS

**Charger/USB insertion and extraction.** In this task, the robot is required to remove a USB connector or a charger from its original socket and subsequently insert it into the correct one. We define task success only when the robot fully completes the entire sequence of actions and applies sufficient force to ensure the plug is firmly seated in the socket; otherwise, the attempt is considered a failure. Compared to the two-pronged charger, the USB connector has a smaller aperture and a more constrained tolerance margin, resulting in a relatively lower success rate.

**Tabletop Grasping.** In this task, the robot receives images of various objects along with textual prompts, and is required to apply different levels of grasping force to lift the objects from the table and place them into a red tray whose location is randomly determined. For each object, the robot must first identify its position and then grasp it with precision. Soft objects must not be deformed during the process, whereas denser objects require sufficient force to ensure a stable grasp. Task success is defined strictly as placing the object securely into the tray; any deviation from this outcome is considered a failure.

**Wiping the Board.** In this task, the robot is required to grasp a blackboard eraser with sufficient normal and tangential force, and then drag it to erase the writing on the board. The primary challenge lies not only in the successful execution of the wiping action itself, but also in the robot's ability to assess task success in real time, in order to decide whether additional wiping is necessary and how the applied force should be adjusted. The task is considered successful only if, upon completion of the wiping motion, all writing on the blackboard has been removed, the eraser is returned to its original position, and no excessive redundant attempts have been made. Otherwise, the task is regarded as a failure.

## B  HARDWARE SETUP

Our primary platform is a single 7-DoF Franka arm equipped with a Weiss WSG-50 parallel-jaw gripper. A wrist-mounted GoPro camera with fisheye lens provides wide-angle observations. The arm is mounted on a custom height-adjustable table that can be pushed by a person—while not autonomous, this mobility allows us to evaluate the policy beyond traditional laboratory environments. The action space is 7-dimensional (6-DoF end-effector pose plus gripper width). Expert demonstrations for this platform are collected using UMI described in Section 2.4.

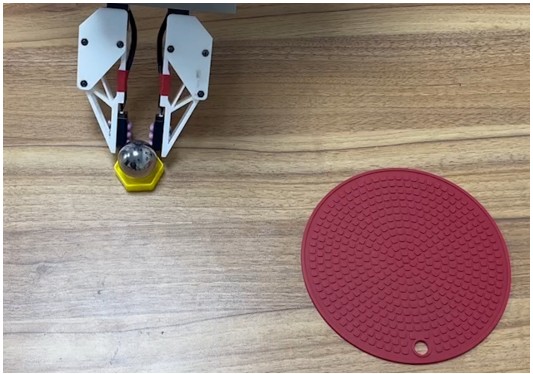

Figure 8: Failure in Tabletop Grasping task.

## C  FAILURE CASES

Despite the promising performance of Tactile VLA, it still makes mistakes under certain circumstances. As shown in the Figure 8, under adverse lighting conditions the robot may fail to accurately localize the target object, thereby being unable to successfully complete the pick-and-place task. Similarly, for tasks requiring fine-grained manipulation, such as USB insertion, although Tactile VLA achieves a substantially higher success rate than the baselines, it remains limited to only 35%.

In most cases, the robot struggles to perform mid-air alignment and insertion into the correct socket, ultimately resulting in task failure. We firmly believe that training Tactile VLA on a larger-scale robotic dataset will yield significantly higher success rates across these challenging tasks.

## D    DATA COLLECTION SETUP

For our tactile-aware UMI, we carefully considered the problem of temporal synchronization. Before each collection session, we align the timestamps across all data streams. During collection, we capture 100Hz tactile feedback and 20Hz visual data, then subsequently down-sample the high-frequency tactile signals to match them with their corresponding visual frames. The resulting VLA-T training dataset contains precisely synchronized multimodal information from vision, language, tactile sensing, and action trajectories.

