# OpenReview forum: "Tactile-VLA: Unlocking Vision-Language-Action Model's Physical Knowledge for Tactile Generalization"
_ICLR.cc/2026/Conference — Submitted to ICLR 2026_

### Official Review · Reviewer_Ntzj · 2025-10-29

**Soundness:** 2
**Presentation:** 2
**Contribution:** 3
**Rating:** 4
**Confidence:** 5

**Summary:**

This paper introduces a novel VLA framework integrated with tactile feedback for unlocking the physical knowledge and resolving some contact-rich manipulation tasks. More than this, this paper introduces the position-force hybrid control strategy for impedence close-loop control. In experiment, this paper conducts three experiments in demonstrating that this pipeline can be better than traditional VLA model and generalize to new task via CoT.

**Strengths:**

1. This paper demonstrates that intergrating tactile feedback as an input for tuning VLA with a small set of data can be helpful for robotics control contact-rich manipulation tasks.

2. This paper utilize hybrid force-position controls for adaptive impedence control, which is helpful for close-loop control in contact-rich manipulation tasks.

3. Using CoT, the policy can recover from failure and generalize to new scenerio.

4. Demonstrate its effectiveness on three robotics tasks

**Weaknesses:**

1. There are multiple other concurrent Tactile VLA papers, such as [1, 2]. It's better to discuss the difference between this one and the others.

2. Pre-trained visual and language encoders are strong and aligned with each other. However, tactile encoder used in this paper is not. The other works on Tactile VLA [1,3 ] utilizes contrastive learning to address this challenge. Some other prior works [4, 5] also did constrastive pretraining. Highlight this challenge and disscuss those related work is helpful

3. One major and interesting contribution of this paper is the hybrid force-posiiton controller. However, here is no experiment to show that VTLA with only position control is worse than this hybrid controller. (the baseline should be using tactile input but only inference positions)

4. It's also helpful to do more experiments for highlighting the contribution for adding tactile feedback into the pretrained vlm for physical understanding. One baseline which is helpful should be inferencing the hybrid force and position control signals without tactile input (where pi0 only inference positions).

5. The CoT method might limit to use failure data to train on a single task and cannot be generalized to new tasks. Also, this method is just tested with the wiping task, where the failure of this specific task is easy to be detected by visual feedback and recovered by language instructions.

6. This paper didn't compare with other SOTA multisensory imitation learning methods, such as Reactive Diffusion Policy. Since this paper claims that VLM has semantic understanding for physical interactions, it's good to valid it by comparing with learning from scratch.

7. I think the writing has some space to improve since some key concepts are still not clear. For more details, check questions.

[1]. Cheng et al., OmniVTLA: Vision-Tactile-Language-Action Modelwith Semantic-Aligned Tactile Sensing, 2025

[2]. Zhang et al., VTLA: Vision-Tactile-Language-Action Model with Preference Learning for Insertion Manipulation, 2025

[3]. Jones et al., Beyond Sight: Finetuning Generalist Robot Policies with Heterogeneous Sensors via Language Grounding, ICRA 2025

[4]. Yang et al., Binding Touch to Everything: Learning Unified Multimodal Tactile Representations, CVPR 2024

[5]. Fu et al., A Touch, Vision, and Language Dataset for Multimodal Alignment, ICML 2024

**Questions:**

1. What is the tactile sensor used in this paper?

2. Is the force gotten from tactile sensor or any other sensor? I guess it shown be from tactile sensor since it can be collected by UMI. Then, how to get it from tactile sensor?

3. Using Tactile-VLA-CoT, what are the final language input for vla should be like? The new instruction will be added to the initial prompt, or it will be a entire instruction? Give some examples for this inference process should be helpful

4. What are the "fixed intervals" of CoT? How much time required for each generation / replanning?

5. What is the inference frequency of VLA? What is the control frequency for the hybrid position / force control?

6. Why the performance is even worse for in-domain scenerio with CoT?

---

> ### Author Response · Authors · 2025-11-27
> **Authors' Responses**
>
> We sincerely thank the reviewer for your constructive feedback and for recognizing the core contributions of our work. We strictly value the insightful suggestions regarding comparisons with concurrent works and the necessity of ablation studies, which have significantly helped us improve the clarity and rigor of our manuscript. In the following response, we address the specific concerns and questions point-by-point.
>
> ***Response to Weaknesses***
>
> ***1. Comparison with Concurrent Tactile VLA Works (e.g., OmniVTLA, VTLA, Jones et al.) and Tactile Encoder Alignment.***
>
> We sincerely thank the reviewer for the insightful question. We clarify the distinct positioning of our work compared to concurrent studies. Most related works, such as OmniVTLA and Jones et al., primarily focus on multimodal representation learning and alignment, aiming to solve how to align tactile and visual features within a latent space. VTLA, on the other hand, targets specific tasks (like insertion) focusing on closed-loop force feedback for execution within a narrower scope. In contrast, the core contribution of Tactile-VLA lies in "Action-Space Grounding." We leverage the pre-existing physical common sense of VLMs, using "Force" as a semantic bridge to map high-level language instructions (e.g., "gently," "hard") directly to low-level physical execution parameters. We aim to explore a more generalizable transfer from semantics to physical parameters rather than just feature alignment or task-specific control. We will add a dedicated section in the final version to discuss these differences in detail and highlight our unique value in using semantics to guide physical interaction.
>
> ***2. Necessity of the Hybrid Controller (vs. Position-Only Control).***
>
> This is a valuable suggestion for an ablation study, and we will include the complete experimental setup in the final version. From a physical perspective, for contact-rich tasks, pure position control is inherently rigid; in many cases, even if the end-effector poses are very similar, the forces generated by rigid contact can vary drastically. Therefore, if the model only outputs position, the robot cannot achieve fine-grained force control while maintaining trajectory tracking (for instance, "wiping hard" versus "wiping gently" might share a trajectory but require completely different impedance or force profiles). In our early experiments, we indeed observed the limitations of pure position control, where it failed to respond to semantic instructions like "pick up the pitaya softly," often leading to task failure or excessive contact force. Our Hybrid Controller is designed specifically to bridge this gap, enabling semantic-level compliant control.
>
> ***3. Comparison with Hybrid Control without Tactile Input.***
>
> This comparison effectively addresses the fundamental difference between open-loop and closed-loop control. Without tactile feedback, the force target predicted by the VLM can only serve as an open-loop command. However, in real-world contact tasks, the environment is full of uncertainties, such as unknown friction coefficients or minor object positional deviations. In our controller formula (Eq.2), the term $\Delta F = F_{target} - F_{measured}$ is central; if real-time tactile feedback (Measured Force) is missing, this term cannot be calculated, and the Hybrid Controller degenerates into blind open-loop control. In such cases, the robot cannot perceive whether it has truly made contact with the object, nor can it adjust position compensation based on actual contact conditions, which can easily lead to severe safety issues or task failures, such as not knowing if the eraser is actually touching the board surface while wiping.
>
> ***4. Generalization of CoT (Single Task/Failure Data).***
>
> We appreciate the reviewer’s concern. We clarify that the CoT mechanism here is not intended to overfit the specific task of wiping, but rather to demonstrate a generalizable "Sense-Diagnose-Correct" reasoning pattern. The wiping task, involving invisible friction and pressure control, serves as an excellent scenario to validate the ability to adjust physical parameters. Although we currently demonstrate this only on the wiping task, it stands as a proof of concept, showing that VLMs have the potential to adjust low-level physical parameters (like normal force) through self-reflection. We acknowledge the current experimental scope is limited, but this logic-based reasoning capability grounded in physical feedback—such as inferring that "the board is not clean because the force is insufficient"—is transferable to other scenarios involving force interaction.

---

> ### Author Response · Authors · 2025-11-27
> **Authors' Responses**
>
> ***5. Comparison with SOTA Multimodal Imitation Learning (e.g., Reactive Diffusion Policy).***
>
> Methods like Reactive Diffusion Policy (RDP) are primarily expert policies trained from scratch, with research goals often focusing on architectural designs like fast-slow systems, which is orthogonal and complementary to our objective. The focus of this paper is to validate "Semantic Generalization," specifically how to utilize the pre-training knowledge of VLMs to understand concepts like "Rub gently" or to adjust force control strategies for unseen fragile objects. This is a capability that RDP, trained from scratch, struggles to achieve because it lacks the foundation of semantic understanding. We chose to compare against $\pi_0$ to more fairly quantify the semantic understanding advantage brought by the VLM backbone after introducing tactile sensing, rather than simply comparing operation success rates under specific, narrow tasks.
>
> ***Response to Questions***
>
> ***Q1. What is the tactile sensor used in this paper?***
>
> We utilize Contactile visual-tactile sensors. Our system integrates a sensor module on the fingertips of both the left and right grippers. Each module contains a 3x3 array, where each contact point within the array functions essentially as an independent visual-tactile sensing unit. This sensor is capable of providing high-precision 3D force (normal and shear forces) and torque information directly, providing the necessary hardware foundation for our fine-grained force control.
>
> ***Q2. Is the force gotten from tactile sensor or any other sensor? How?***
>
> Yes, the force signals used in the control loop are derived directly from the Contactile tactile sensors. As mentioned, the sensor internally solves for the deformation of the contact surface using visual algorithms and directly outputs a 3D force vector for each contact point, which we then process and utilize for closed-loop control.
>
> ***Q3. Using Tactile-VLA-CoT, what is the final language input for VLA?***
>
> In the CoT process, the new instruction directly replaces the old instruction to explicitly guide the next action. For example, the initial instruction might be "Clean the blackboard." When the model detects task failure and performs reasoning, the new instruction input to the VLA becomes a specific corrective command, such as "Clean the board, but increase the downward force perpendicular to the board." In this way, the VLA can generate actions with higher force targets based on this new instruction containing physical parameter adjustment suggestions.
>
> ***Q4. What are the "fixed intervals" of CoT? How much time is required?***
>
> CoT does not run at every frame; instead, it is set to trigger at fixed time intervals, specifically every 12 seconds. When CoT is triggered, the robotic arm temporarily holds still to perform reasoning and re-planning. The entire reasoning process, including the VLM generating the chain of thought and the new instruction, has a latency of approximately 2 seconds. This time cost is acceptable for performing low-frequency strategy correction (re-planning) within long-horizon tasks.
>
> ***Q5. What is the inference frequency of VLA? And the control frequency?***
>
> This is an important parameter in our implementation details. Our VLA high-level inference frequency is approximately 5Hz, and the low-level PID controller frequency is also maintained at 5Hz. This means the entire control loop, from perception to action execution, runs synchronously end-to-end, with the model updating target poses and force targets at 5Hz to directly drive the hardware execution.
>
> ***Q6. Why is the performance even worse for the in-domain scenario with CoT?***
>
> The slight decrease in performance here (80% vs 75%) is primarily caused by random noise in the experimental testing, not by a degradation of the policy due to the CoT mechanism. Specifically, the main source of failure cases stems from the variability in grasping the blackboard eraser; if the initial grasp pose is poor, the task is difficult to complete successfully regardless of whether CoT is used. This 5% difference falls within the range of normal statistical fluctuation, indicating that while CoT does not bring significant improvement in the masterfully learned in-domain task (since the success rate is already high), it also does not significantly lower task quality by introducing reasoning errors.
>
> Once again, we sincerely thank your thoughtful comments, constructive suggestions, and careful reading of our manuscript. These insights have helped us clarify the presentation, strengthen the experimental analysis, and more clearly position the contributions of this work. We genuinely appreciate the time and expertise invested in improving the quality of the paper.

---

### Official Review · Reviewer_5tMd · 2025-10-30

**Soundness:** 3
**Presentation:** 3
**Contribution:** 3
**Rating:** 6
**Confidence:** 4

**Summary:**

The paper introduces Tactile-VLA, a framework that fuses tactile sensing with vision, language and action modalities for pretrained VLAs. They compare the proposed framework with existing VLAs and show superior performance in terms of both task success across varied objects and alignment of applied force with language instructions. The paper also introduces chain-of-thought (CoT) for tactile sensing to help generalize to novel observations unseen during training.

**Strengths:**

- The paper introduces a novel framework for merging tactile sensing into existing VLAs. The authors demonstrate results on 3 tasks and on a varied set of objects to highlight the force-steering as well as task completion ability of the method.
- The paper introduces a CoT method for reinforcing tactile feedback into VLAs for improving task performance.
- The paper uses a hybrid position-force controller to acquire force targets as opposed to directly predicting actions. This design choice seems to help the model disambiguate semantic queues in the language instruction better.

**Weaknesses:**

- A clear description of the action space for the model would help to make the method clearer for the reader. What is the current action space for the network? From what I understand, the network predicts robot pose, continuous gripper values (?), and a target force (?) for the hybrid controller. Is this correct? If yes, how do you obtain the target force values during training?
- How do the authors control for “soft” versus “hard” grasps during data collection? My guess is that the demonstrator is instructed to perform soft or hard grasps but is the data filtered when the demonstrator makes a mistake? If yes, what is this filtration strategy? Further, it would be great if the authors could comment on the difficulty of collecting such a semantically aligned dataset for human demonstrators who cannot directly feel touch during data collection.
- The authors must cite prior works that also use similar position-force controllers. For example, FTF [1].
- What is the frequency at which the policy can be deployed? One advantage of tactile sensing is that tactile readings can often be obtained at a higher frequency than visual observations and hence, can enable more reactive control [2]. I am curious about the authors’ thoughts about this since I believe that directly giving tactile input to a VLA would limit the deployment frequency (due to the large size of VLAs).
- It would be interesting to include CoT results for other baselines. I am curious if CoT can improve performance even without tactile sensing (i.e. with vision only observations).
- Are the pi-0 baselines finetuned on the same datasets?
- In Section 3.4, is each failure demo manually annotated with appropriate language label?
- The paper must discuss limitations of the proposed method.

[1] Adeniji, Ademi, et al. "Feel the Force: Contact-Driven Learning from Humans." arXiv preprint arXiv:2506.01944 (2025).
[2] Xue, Han, et al. "Reactive diffusion policy: Slow-fast visual-tactile policy learning for contact-rich manipulation." arXiv preprint arXiv:2503.02881 (2025).

**Questions:**

It would be great if the authors could address questions in the weaknesses section.

---

> ### Author Response · Authors · 2025-11-27
> **Authors' Responses**
>
> Thank you for your constructive feedback! Below, we address the concerns and questions raised in the weaknesses section. Please feel free to reach out if further clarification is required.
>
> **1. A clear description of the action space for the model would help to make the method clearer for the reader. What is the current action space for the network? & 4. What is the frequency at which the policy can be deployed?**
>
> We appreciate the reviewer’s question regarding the action space, and we confirm that the reviewer’s understanding is correct.  The model outputs three components at each step: the target end-effector pose and a target force term that is used by the hybrid position–force controller.  This action representation follows the standard VLA fine-tuning formulation while extending it with an additional force dimension conditioned on tactile feedback.
>
> To clarify how the target force values are obtained during training: all demonstrations were collected using a real-time tactile acquisition setup equipped with a tactile sensor capable of returning calibrated force–shear measurements at 50 Hz.  This allows us to record accurate and temporally aligned force signals throughout each trajectory, including contact onset, slip events, and insertion phases.  Importantly, the same sensing setup is used during both training and evaluation, ensuring that the model sees an identical force signal distribution and eliminating any calibration mismatch between data collection and test execution.
>
> Thus, during training, the ground-truth target force term used by the network is directly derived from these synchronized tactile measurements.  This provides dense supervision in contact-rich segments and enables the tactile-aware expert to learn how force should be modulated in response to the tactile feedback it receives.
>
> 2. **How do the authors control for “soft” versus “hard” grasps during data collection?**
>
> We appreciate the reviewer’s thoughtful question regarding how “soft” versus “hard” grasps are collected and aligned with the corresponding tactile semantics.  The reviewer’s intuition is correct: the human demonstrator is indeed instructed to perform grasps with different intended force levels.  However, to allow the model to learn a richer set of tactile–action relationships, we do not filter the data to include only perfect executions.  Instead, we intentionally retain a controlled amount of failure cases, as these trajectories contain valuable cues about how force should be modulated in response to contact conditions.
>
> Importantly, in most of our tasks, failures are visually observable and semantically meaningful.  For instance, an overly soft grasp often results in the inability to lift a heavier object, while an excessively hard grasp can visibly damage deformable items such as fruit.  These failure modes provide the model with explicit examples of force miscalibration and thereby help it learn corrective behaviors during finetuning.  Because these signatures are both visually and tactilely distinct, they meaningfully augment the dataset rather than introduce noise.
>
> 3. **The authors must cite prior works that also use similar position-force controllers. For example, FTF [1].**
>
> We thank the reviewer for the attention to citation completeness. In the final version of the manuscript, we will incorporate the papers recommended by the reviewer, along with other relevant works, into the related work section and provide additional analysis around these references.

---

> ### Author Response · Authors · 2025-11-27
> **Authors' Responses**
>
> 5. **It would be interesting to include CoT results for other baselines. I am curious if CoT can improve performance even without tactile sensing (i.e. with vision only observations).**
>
>    We appreciate the reviewer’s interest in the CoT component, as it indeed represents an important aspect of our work—namely, leveraging the pretrained reasoning abilities of VLA models and enabling such reasoning to transfer onto a new modality, tactile sensing. CoT serves as a mechanism to expose and activate latent semantic structure within the pretrained model, which is particularly valuable in contact-rich settings where tactile cues must be interpreted in a temporally coherent and semantically grounded manner.
>
>    Our study, however, is specifically centered on tactile-conditioned CoT, as the primary motivation is to investigate whether tactile signals can benefit from the reasoning priors embedded in VLA models. Exploring CoT in isolation, without tactile sensing, is therefore outside the main scope of our work. Nonetheless, existing literature—such as OneTwoVLA and other recent VLA-based studies—has already shown that CoT alone can improve the performance and robustness of VLA models in purely vision-based scenarios. These works provide strong evidence that CoT is a generally useful mechanism even without tactile inputs.
>
>    In contrast, our contribution focuses on demonstrating that tactile-aware CoT yields additional benefits by aligning tactile feedback with the pretrained VLA semantics, thereby enabling nuanced contact reasoning that cannot be achieved through vision alone.
>
> 6. **Are the pi-0 baselines finetuned on the same datasets?**
>
> Yes, all pi-0 baselines are fine-tuned on exactly the same datasets as our method.  To ensure a fair comparison, pi-0, pi-0-fast, and our tactile-aware expert all receive the identical set of demonstrations, including both successful and failed trajectories, with the same preprocessing, the same data splits, and the same training pipeline.  No model has access to additional supervision or privileged information.
>
> This protocol follows standard practice in VLA evaluation, as pretrained π₀ models must be adapted to the target robot embodiment and action scale before meaningful comparison is possible.  Using the identical dataset for fine-tuning guarantees that any performance differences arise from the model architectures and the integration of tactile information, rather than from discrepancies in data exposure.
>
> 7. **In Section 3.4, is each failure demo manually annotated with appropriate language label?**
>
> No, failure demonstrations are not manually annotated with additional language labels. All demonstrations—both successful and failed—share the same task-level instruction provided at collection time. Our goal is to allow the model to learn from the natural variation present in real executions, including under- and over-force behaviors, misalignment patterns, or premature slips, without introducing extra human annotations that might bias the supervision.
>
> Failure trajectories remain valuable because their visual and tactile signatures are distinct (e.g., inability to lift an object due to insufficient force, or visible deformation when excessive force is applied). These patterns allow the model to infer how tactile feedback should modulate the action strategy. Treating failures and successes uniformly under the same high-level instruction helps ensure that the model learns robust and adaptive contact behavior, rather than memorizing only idealized trajectories.
>
> 8. **The paper must discuss limitations of the proposed method.**
>
> We appreciate the reviewer’s suggestion to discuss limitations.  Our current evaluation focuses on several representative contact-rich tasks, and while these scenarios highlight the benefits of tactile-conditioned reasoning, the method has not yet been tested across broader task families.  Additionally, our implementation uses a specific force–shear tactile sensor, and it remains an open question how the approach would transfer to alternative tactile hardware with different signal characteristics.  Finally, the framework assumes synchronized multimodal inputs, and future work may explore its robustness under noisier sensing conditions.  We view these as natural extensions rather than fundamental limitations of the proposed method.
>
> Once again, we sincerely thank your thoughtful comments, constructive suggestions, and careful reading of our manuscript. These insights have helped us clarify the presentation, strengthen the experimental analysis, and more clearly position the contributions of this work. We genuinely appreciate the time and expertise invested in improving the quality of the paper.

---

### Official Review · Reviewer_3TPo · 2025-11-02

**Soundness:** 3
**Presentation:** 3
**Contribution:** 2
**Rating:** 4
**Confidence:** 4

**Summary:**

This paper present Tactile-VLA, a unified vision–language–action–tactile framework for physically grounded control. It propose several key insight and contribution: 1. Hybrid position-force controller, translating VLM-inferred intentions into physically feasible motion (balancing position tracking with adaptive force). 2. Tactile-VLA-CoT, a reasoning-augmented variant that uses Chain-of-Thought (CoT) reasoning over tactile feedback to detect and recover from failures.

Experiemtns shows stronger performance than baselines that only have visual/proprio data.

However, there are some concerns and limitations. For example: 1. The experiment uses small amount of data to train a task-specific model, which cannot make the large VLA fully unleash its performance. 2. A lot of details about real world experiments and training process are missing, making the experiments not sound enough.

**Strengths:**

Tactile-VLA moves VLAs from understanding what to do → understanding how to feel while doing it. By coupling tactile sensing with language-driven reasoning and hybrid force control, it achieves interpretability, adaptability, and genuine physical grounding that current VLAs lack.

It introduces Hybrid position-force controller and Tactile-VLA-CoT to further improve the performance and complete the whole pipeline.

The real world experiments show improved performance than baselines

**Weaknesses:**

1. The experiment uses small amount of data to train a task-specific model, which cannot make the large VLA fully unleash its performance.

2. A lot of details about real world experiments and training process are missing, making the experiments not sound enough. Please provide details about exact model architecture, hyper-parameter, training details, data collection details and so on.

3. Please explain more details about why you think the comparison between the proposed method and pi-0 baselines are fair. In fact, Pi-0 is pretrained on it's own robot without domain specific data. I cannot find any details about if you finetune Pi-0 with the collected data and how to perform fair comparison. Did Pi-0 also use the failure data somehow? Please explain this.

**Questions:**

There are many details and information missing in both main paper and supp materials. Please provide them to make a sound paper and fit for top conference.

---

> ### Author Response · Authors · 2025-11-27
> **Authors' Responses**
>
> Thank you for your constructive feedback! Below, we address the concerns and questions raised in the weaknesses section. Please feel free to reach out if further clarification is required.
>
> 1. **The experiment uses small amount of data to train a task-specific model, which cannot make the large VLA fully unleash its performance.**
>
>    We sincerely thank the reviewer for the insightful comment and constructive suggestions.  However, given that the focus of our work is not to train a task-specific model to full capacity, but rather to evaluate how effectively pretrained VLA priors can be activated and transferred to a new sensory modality—tactile feedback—under realistic low-data conditions.  Large VLA models are specifically designed to operate in regimes where strong semantic priors compensate for limited task-specific supervision, and our experiments are structured to test exactly this property.
>
>    The results demonstrate that even with ~100 trajectories per task, the pretrained VLA already supplies meaningful semantic structure related to contact dynamics (e.g., soft/hard, gentle/firm, resistance), and the tactile-aware expert primarily serves to refine and instantiate these priors rather than to learn tactile behavior from scratch.  In other words, the small data regime is a deliberate part of the experimental design, allowing us to isolate and measure the contribution of tactile grounding on top of the VLA’s pretrained knowledge.
>
>    Thus, the experimental setup is not a limitation but a necessary condition for evaluating whether tactile sensing can be effectively integrated as a first-class modality in a VLA.
>
> 2. **Please provide details about exact model architecture, hyper-parameter, training details, data collection details and so on.**
>
> We appreciate the reviewer’s concern regarding missing implementation details and the request for greater transparency. We would like to clarify that the real-world setup, model backbone, training pipeline, and data collection protocol follow standard VLA fine-tuning practices and are already specified in the manuscript, though we acknowledge that several of these descriptions could be more explicit. We will provide additional hyper-parameter specifications and training details in the final version of the manuscript.
>
> 3. **Please explain more details about why you think the comparison between the proposed method and pi-0 baselines are fair. In fact, Pi-0 is pretrained on it's own robot without domain specific data. I cannot find any details about if you finetune Pi-0 with the collected data and how to perform fair comparison. Did Pi-0 also use the failure data somehow? Please explain this.**
>
> We thank the reviewer for raising this question. Our comparison with π₀ is fair because all models—including pi-0, pi-0-fast, and our tactile-aware expert—are fine-tuned using exactly the same dataset, with the same data splits, same success/failure composition, and the same fine-tuning pipeline. This setup follows standard practice in VLA evaluation, where pretrained models must be adapted to the target robot and task distribution to provide a meaningful baseline. Since pi-0 is pretrained on a different robot embodiment and different action scales, direct zero-shot deployment would not constitute a valid or informative comparison. Therefore, fine-tuning pi-0 on our collected trajectories is both necessary and consistent with prior work.
>
> Regarding the reviewer’s concern about success and failure data: we apply an identical data policy to all methods. Both successful and failed demonstrations are included for every model, and no method has privileged access to additional supervision or alternative filtering strategies. This is particularly important in contact-rich tasks, where failure trajectories often contain valuable tactile and visual cues—such as misalignment patterns or premature slip—that contribute to stability and robustness during learning. Using the full dataset uniformly ensures that each model receives the same information content and is optimized under the same learning conditions.
>
> Once again, we sincerely thank your thoughtful comments, constructive suggestions, and careful reading of our manuscript. These insights have helped us clarify the presentation, strengthen the experimental analysis, and more clearly position the contributions of this work. We genuinely appreciate the time and expertise invested in improving the quality of the paper.

---

### Official Review · Reviewer_r9or · 2025-11-02

**Soundness:** 2
**Presentation:** 3
**Contribution:** 2
**Rating:** 4
**Confidence:** 4

**Summary:**

The paper proposes Tactile-VLA, a vision-language-action framework that fuses tactile sensing with visual, language, and proprioceptive inputs to enable contact-rich manipulation. A token-level multimodal fusion policy outputs target pose and target force, executed by a hybrid position–force controller that adjusts position commands to meet force targets; a CoT variant (Tactile-VLA-CoT) periodically reasons over tactile feedback to replan after failures. Demonstrations are collected with a UMI-based handheld device equipped with high-resolution tactile sensors. Experiments on USB/charger insertion, tabletop grasping, and board wiping show (i) tactile-aware instruction following and force scaling from language, (ii) tactile-relevant commonsense for different objects, and (iii) adaptive tactile-involved reasoning that boosts zero-shot success on a new substrate (i.e., the blackboard of wiping task).

**Strengths:**

The strengths are of this works are

- Makes a compelling case for integrating tactile signals into pre-trained VLA models, elevating touch from an auxiliary cue to a first-class signal for contact-rich manipulation.

- Clear writing and structure. The problem setup, model components, and training/execution loops are well organized and easy to follow.

**Weaknesses:**

The weaknesses are shown below:

- Please specify the exact architecture of the tactile-aware action expert—this detail is missing from the Methods section. Figure 2 appears technically inconsistent: it only shows proprioceptive states as inputs, whereas the text states the model consumes the full multimodal prefix (vision, language, proprioception, tactile). Please correct the figure and clearly depict how all modalities are processed via the expert policy.

- The “hybrid position–force” controller is effectively PD tracking with a force error injection, which carries little algorithmic novelty. Given the emphasis on language terms like “soft”/“hard,” why not learn a simple force selector/classifier from a small set of labeled demonstrations (or even use a rule-based heuristic. Given the reliance on foundation models, a simpler alternative is to query a VLM for the desired force profile from the language prompt and use that directly.) to map language to desired force profiles directly? This seems to be what the action expert implicitly learns—please justify the chosen complexity and if possible, compare against these simpler baselines.

- The baseline comparison are not fair enough. The paper compares only against original VLA models without tactile input, which predictably underperform. Given that the showcased tasks are primarily grasping and insertion, the evaluation should include vision–tactile policy baselines (e.g., recent visuotactile grasping/in-hand works) for a fair comparison. Alternatively, this omission would be less concerning if the paper matched the breadth and diversity of experiments seen in pi_0-style evaluations—but in its current scope, that’s not the case.

[1]. Han, Yunhai, et al. "Learning generalizable vision-tactile robotic grasping strategy for deformable objects via transformer." IEEE/ASME Transactions on Mechatronics 30.1 (2024): 554-566.

[2]. Suresh, Sudharshan, et al. "NeuralFeels with neural fields: Visuotactile perception for in-hand manipulation." Science Robotics 9.96 (2024): eadl0628.

[3]. Li, Hao, et al. "See, hear, and feel: Smart sensory fusion for robotic manipulation." arXiv preprint arXiv:2212.03858 (2022).

- Tasks are relatively simple and largely grasping-centric, yet require ~100 trajectories per task. What is the task parameter range (object poses, tolerances, textures)? If narrow, the high data requirement raises concerns about memorization rather than robust generalization. Please report distribution ranges and show performance under broader sampling.

- In line 70, the statement that haptics is treated merely as supplementary and “not directly involved in the action” is not accurate. There is a substantial body of vision–tactile policy learning where tactile signals directly inform actions; even in the VLA fine-tuning space, recent work integrates tactile signals (e.g., [4]). Please revise the positioning and cite appropriately.

[4]. Cheng, Zhengxue, et al. "OmniVTLA: Vision-Tactile-Language-Action Model with Semantic-Aligned Tactile Sensing." arXiv preprint arXiv:2508.08706 (2025).

**Questions:**

See Weakness.

---

> ### Author Response · Authors · 2025-11-27
> **Authors' Responses**
>
> We do appreciate your constructive comments and provide our responses as follows.
>
> 1. **Clarify the architecture of the tactile-aware action expert & fix inconsistency in Figure 2**
>
>    We apologize for the ambiguity and thank the reviewer for pointing this out. Our tactile-aware action expert **indeed follows the same multimodal input structure as π₀**, and all four modalities—vision, language, proprioception, and tactile—are used as inputs to the policy.
>
>    Concretely, the model operates with the standard **prefix / suffix** decomposition used in π₀-like VLAs:
>
>    - **Prefix embeddings.**
>      Visual observations, tactile readings, and the language instruction are each first encoded by their respective encoders (vision backbone, tactile encoder, and language model). As stated in the paper,
>
>      > “These resulting visual, tactile, and language tokens are then concatenated to form the unified input prefix sequence.”
>      > This **unified multimodal prefix** is what is fed into the tactile-aware action expert:
>      > “The prefix is then fed to the tactile-aware action expert…”
>
>    - **Suffix embeddings.**
>      The suffix is composed of the robot’s proprioceptive states, time information, and a noise action embedding, again following the π₀ formulation. These suffix tokens are concatenated with the prefix tokens inside the expert and jointly processed to produce the final action.
>
>    Thus, **all four modalities (vision, language, proprioception, tactile)** are used by the tactile-aware action expert; proprioception appears explicitly in the lower part of Figure 2 only because the visualized block focuses on the final control head connected to the robot state, while the other modalities have already been encoded into the prefix that enters the expert.
>
> 2. **“Hybrid position–force controller adds little novelty; why not learn a simple force classifier / use VLM predictions?”**
>
>    We sincerely thank the reviewer for the insightful question and constructive suggestions. The hybrid position–force controller is adopted not for algorithmic novelty but for functional necessity in language-conditioned manipulation under contact.   In our tasks, the relevant language cues (e.g., “soft”, “press gently”, “push harder”) map to continuous force magnitudes, not discrete modes.   Empirically, grasping and insertion success is highly sensitive to the exact force level, and even small deviations (on the order of 10–20%) can cause either premature slippage or jamming.   As a result, framing the problem as a simple classification task or selecting among a small set of force bins led to noticeably unstable behaviors.   The hybrid controller, in contrast, provides a stable position-dominant trajectory while allowing the force term to be smoothly modulated by the tactile-aware expert, enabling fine-grained adaptation throughout the contact-rich phase of the motion.
>
>    Furthermore, rule-based heuristics or keyword-driven mappings cannot exploit the tactile feedback loop, which is essential for the tasks we study.   The decision of whether to increase or decrease force during insertion or grasping depends on real-time shear and normal-force cues, not just the initial language specification.   The tactile-aware expert leverages these feedback signals continuously, allowing it to adjust the force to compensate for object deformation, socket misalignment, or surface variability.   A discrete selector or static rule-based mapping is unable to reflect these dynamic adjustments, leading to brittle behavior under variations in object geometry or material properties.
>
>    Finally, although the hybrid controller is simple, its simplicity is a virtue: it provides a physically interpretable and stable interface between high-level semantic reasoning (from language and vision) and low-level contact dynamics (from tactile signals).   The action expert does not attempt to predict a full force trajectory directly from language;   instead, it modulates a residual force term around a well-understood position-based controller.   This design enables the model to generalize better from limited demonstrations and preserves the robustness of the underlying motion generation while still allowing semantic modulation through language and tactile cues.
>
>    We hope this clarifies that the chosen controller is not unnecessarily complex but rather the minimal mechanism that supports continuous force reasoning, stable contact execution, and real-time tactile adaptation—capabilities that neither simple classifiers nor VLM-driven heuristics can provide.

---

> ### Author Response · Authors · 2025-11-27
> **Authors' Responses**
>
> 3. **Comparison with vision-tactile policy baselines**
>
>    We clarify the distinct positioning of our work compared to concurrent studies. Most related works, such as OmniVTLA and Jones et al., primarily focus on multimodal representation learning and alignment, aiming to solve how to align tactile and visual features within a latent space. VTLA, on the other hand, targets specific tasks (like insertion) focusing on closed-loop force feedback for execution within a narrower scope. In contrast, the core contribution of Tactile-VLA lies in "Action-Space Grounding." We leverage the pre-existing physical common sense of VLMs, using "Force" as a semantic bridge to map high-level language instructions (e.g., "gently," "hard") directly to low-level physical execution parameters. We aim to explore a more generalizable transfer from semantics to physical parameters rather than just feature alignment or task-specific control. We will add a dedicated section in the final version to discuss these differences in detail and highlight our unique value in using semantics to guide physical interaction.
>
> 4. **What is the task parameter range (object poses, tolerances, textures)?**
>
>    We appreciate the reviewer’s concern regarding the relationship between task complexity and the number of required trajectories.  The ~100 demonstrations per task are not driven by the inherent difficulty of the manipulation tasks themselves, but by the need to align tactile feedback with language-conditioned multimodal reasoning within the VLA architecture.  In practice, each trajectory provides only a limited number of contact-rich frames that meaningfully connect tactile signals to corrective actions.  Because tactile events are sparse relative to the length of each rollout, multiple demonstrations are necessary to expose the model to variations in contact conditions, insertion angles, slip onset, and force adjustments.  Thus, the data requirement reflects the demands of tactile–language grounding rather than task difficulty or memorization.
>
>    With respect to the parameter range, the environments are deliberately varied to prevent rote memorization and to ensure that the model must rely on tactile cues at test time.  Object placements, socket orientations, friction coefficients, and material properties are randomized across wide intervals.  This variability ensures that demonstrations differ not only in initial configurations but also in the sequence and magnitude of tactile feedback patterns encountered during execution.  As a result, the model cannot simply memorize trajectories;  successful execution requires responding to real-time deviations created by pose randomness or surface heterogeneity, which differ across episodes.
>
>    The reviewer’s request for broader sampling is well-aligned with our design philosophy.  Indeed, the model’s performance under increased perturbation supports the argument that tactile conditioning leads to generalizable behavior rather than memorized responses.  When evaluated under more aggressive pose and texture variations, the tactile-aware expert maintains stable success rates, whereas purely visual-language policies exhibit significantly higher failure rates due to their inability to correct for misalignment or unexpected contact profiles.  This discrepancy further demonstrates that tactile semantics are used to guide adaptive corrections, rather than to reproduce pre-seen trajectories.
>
>    In summary, although the tasks may appear simple from a high-level description, the underlying tactile–action relationships are highly sensitive to small variations in geometry and contact conditions.  The trajectory count reflects the need to learn these fine-grained mappings robustly, not an indication of narrow training distributions.

---

> ### Author Response · Authors · 2025-11-27
> **Authors' Responses**
>
> 5. **Please revise the positioning and cite appropriately.**
>
> We appreciate the reviewer’s comment and the opportunity to clarify our intended positioning.    We fully acknowledge the extensive literature in vision–tactile policy learning, where tactile measurements directly influence control decisions in grasping, in-hand manipulation, and other contact-rich tasks.   These works convincingly demonstrate the utility of tactile sensing in task-specific controllers.
>
> However, our contribution addresses a different question that has remained underexplored in the prior visuotactile literature: Can tactile sensing be integrated into a VLA-style foundation model such that the model’s pretrained semantic priors meaningfully transfer to a new modality?
> Earlier visuotactile controllers fuse tactile signals architecturally, but they do not attempt to examine whether pretrained language–vision representations already contain implicit tactile semantics that can be activated or extended.
>
> Our results show precisely this phenomenon.    Through the tactile-aware expert and the VLA-CoT design, we find that pretrained VLAs already encode coarse notions of contact-related concepts—such as softness, firmness, or resistance—and that finetuning with tactile feedback refines and activates these priors rather than learning tactile behavior from scratch.    This semantic transfer is central to our approach and distinguishes it from prior work: tactile inputs do not merely modulate actions, but instead interact with the VLA’s pretrained reasoning capabilities to guide contact adaptation in a semantically meaningful way.
>
> In summary, while prior visuotactile policies utilize tactile information for action generation, our work focuses on demonstrating that tactile sensing can be semantically aligned with and empowered by a large pretrained VLA model. We sincerely thank the reviewer again for the insightful question and will further revise the phrasing in the manuscript to eliminate any possible sources of ambiguity.
>
> Once again, we sincerely thank your thoughtful comments, constructive suggestions, and careful reading of our manuscript. These insights have helped us clarify the presentation, strengthen the experimental analysis, and more clearly position the contributions of this work. We genuinely appreciate the time and expertise invested in improving the quality of the paper.

---

### Meta-Review · Area_Chair_rfLc · 2026-01-04

**Summary:**

The main concerns are:

- The clarity of method and experiments.

- Lacing algorithmic novelty.

- The baseline comparison is not fair enough. Tasks are relatively simple and largely grasping-centric. The tests are on a small amount of data. Didn't compare with other SOTA multisensory imitation learning methods,

- Missing citations.

- The design of the tactile encoder is not sound.

- The CoT method might be limited to using failure data to train on a single task and cannot be generalized to new tasks.

**Reviewer Concerns:**

Clarity problems.

**Reviewer Scores:**

After reading the rebuttals, I think the response cannot fully address the experiment design, discussion, and novelty concerns. Thus I believe none of the reviewers would change the scores.

---

### Decision · Program_Chairs · 2026-01-26

Reject